# Is Instrumental Compression Equally Effective and Comfortable for Physiotherapists and Physiotherapy Students than Manual Compression? A Comparative Cross-Sectional Study

**DOI:** 10.3390/ijerph182212121

**Published:** 2021-11-18

**Authors:** Sara Pérez-Palomares, Carolina Jiménez-Sánchez, Ignacio Serrano-Herrero, Pablo Herrero, Sandra Calvo

**Affiliations:** 1Department of Physiatry and Nursing, Faculty of Health Sciences, IIS Aragon, University of Zaragoza, 50009 Zaragoza, Spain; saraperez@unizar.es (S.P.-P.); sandracalvo@unizar.es (S.C.); 2Faculty of Health Sciences, Universidad San Jorge, 50009 Zaragoza, Spain; cjimenez@usj.es (C.J.-S.); ignacioserranoherrero@gmail.com (I.S.-H.)

**Keywords:** pressure release, pressure pain threshold, myofascial trigger points, musculoskeletal pain, myofascial pain

## Abstract

The objective of this work is to compare the homogeneity of instrumental and manual compression during the simulation of a pressure release technique, measured with a dynamometer, as well as to evaluate the comparative degree of comfort by physiotherapists and physiotherapy students when performing this technique. Methods: A comparative cross-sectional study was carried out with physiotherapists (lecturers with clinical experience) and 4th year students of the Physiotherapy Degree at Universidad San Jorge. The amount of pressure performed and how it was maintained during 80 s with both techniques was analysed using a digital dynamometer. The degree of comfort was evaluated using a modified numeric rating scale, with higher values representing a higher degree of discomfort. Results: A total of 30 subjects participated. Significant differences were found between the techniques in terms of maintaining a constant pressure level for 80 s (*p* = 0.043). A statistically significant difference was found between both techniques in the period from 45 to 80 s. Regarding the degree of discomfort, the value obtained from the students’ responses was 4.67 (1.35) for the manual technique and 1.93 (0.88) for the instrumental technique. In the case of physiotherapists, the comfort was 4.87 (2.13) for the manual technique and 3.33 (1.54) for the instrumental technique. Conclusion: The sustained manual compression necessary in manual pressure release techniques in the treatment of myofascial trigger points can be performed with assistive tools that guarantee a uniform compression maintained throughout the development of the technique and are more comfortable for physiotherapists.

## 1. Introduction

Manual compression is one of the most commonly used manual therapy techniques in clinical practice by physiotherapists, either alone or included as part of a multimodal treatment [1,2]. This method has proved to be effective in the treatment of myofascial pain provoked by Myofascial Trigger Points (MTrPs) in different conditions [3,4,5]. MTrPs are defined clinically as hyperirritable and tender nodules located within a taut muscle band [6], which may provoke pain and other motor and autonomic dysfunctions [7]. Pressure release techniques consist of a sustained manual pressure [8,9], usually performed with the thumb or fingertip, that is gradually increased until the new tissue barrier [10] according to the therapist’s perception (compression release) or to “comfortable” or “tolerable” pain reported by the patient (ischemic compression) [11]. It is widely recommended to avoid using the term “ischemic compression” and there is agreement on the term “pressure release” (PR) regardless of the method used (therapist perception or patient perception) [8].

Several clinical aspects are considered to be relevant when clinicians apply PR techniques: (1) the site of the application (in the most sensitized location) [12]; (2) the uniformity of compression during the technique application [13,14], and; (3) the duration of compression, which should be between 60–90 s to be effective [15,16]. Related to the last two clinical aspects, the importance of maintaining the necessary pressure during the application time has been demonstrated to be key for the effectiveness in decreasing pain. These studies have been performed by means of a pressure algometer [13,14] not specifically designed as a treatment tool, which results in some degree of discomfort for both the patients and the clinicians if used in the clinical practice [13]. So far, it is unknown if the therapeutic pressure exerted manually by the physiotherapist is kept constant throughout the entire technique when the technique is finger administrated.

Besides, the use of ergonomic tools specially designed to help physiotherapists in their manual techniques could be beneficial given the high prevalence of hand pathology (around 49% wrist and hand injuries, of which 76% related to the thumb) [17], reducing the fatigue or pain related to professional pathologies and making the application of manual treatments easier. Although there have been a few studies evaluating different tools used in PR techniques [18,19] and they have been shown to be effective in decreasing pain [18] and increasing pressure pain thresholds (PPTs) [18,19], no studies have compared how the compression is maintained when performing an instrumental and a manual technique, if there are any differences as the compression time increases, or if there could be any differences between experienced and inexperienced physiotherapists when applying the techniques.

Therefore, the first objective of this work is to compare the capacity to maintain homogeneous pressure by physiotherapists and physiotherapy students when performing a PR technique manually or with an ergonomic tool over a viscoelastic model. The second objective is to evaluate and compare the degree of discomfort that both physiotherapists and physiotherapy students experience when performing these techniques.

## 2. Materials and Methods

### 2.1. Study Design

A comparative cross-sectional study was carried out at the Universidad San Jorge of Zaragoza (Spain). This project has been approved by Universidad San Jorge Ethics Committee (N°005 16/17) and has followed the recommendations of the Declaration of Helsinki of the World Medical Association (WMA) and the Code of Ethics of the Association of Medical Colleges and Physiotherapists of Spain.

All participants were informed of the nature of the study and signed the informed consent document prior to their participation. All the information collected has been treated in accordance with the provisions of Organic Law 15/99 on the protection of personal data.

### 2.2. Participants

Volunteer physiotherapists (lecturers with clinical experience) and students of the 4th year of the Physiotherapy Degree at Universidad San Jorge (USJ) were recruited by email between 17 January and 26 April 2017.

The inclusion criteria for the physiotherapists/lecturers were: (1) being physiotherapist and lecturer in the Physiotherapy Degree at Universidad San Jorge; (2) having clinical experience of at least 3 years performing PR techniques. The only inclusion criterium for students was to be enrolled in the last year (4th) of the Physiotherapy Degree at Universidad San Jorge. The only exclusion criterion for both lecturers/physiotherapists and physiotherapy students was to have any hand and/or fingers pathology at the moment of being recruited (e.g., tendinopathy).

### 2.3. Procedure

The same PR technique was performed manually (PR-Man) or through a device (PR-3T) designed to perform compression on MTrPs (3TOOL^®^ Fisio Consultores SL, Zaragoza, Spain) by both the physiotherapists/lecturers and the physiotherapy students. Both applications of the PR technique were done over a platform which had viscoelastic characteristics, with the aim of imitating the muscle properties. The platform, placed on a bench, was associated with a dynamometer that recorded the pressure exerted during the technique application (80″).

All the participants performed both modalities. The compression techniques were performed sequentially, leaving 4–5 min of rest between one technique and the other. The order in which the subject executed the techniques was previously randomized (1:1 by means of hidden sequence by computer through the online software Research Randomizer) [20] to avoid possible biases.

Before performing the PR techniques, the participants underwent training to define the degree of compression they had to perform manually and with the 3TOOL on the MTrP of the upper trapezius of a healthy volunteer (latent MTrP), so that the participants could have a reference for the amount of pressure that they had to exert over the dynamometer. This value of compression was set up by means of reaching a value of 6 or 7 points out of 10 on the numeric rating scale, or the equivalent “tolerable pain” verbally reported by the volunteer. The reasons for selection of this latent MTrP were that it has a high prevalence, it is superficial and it has shown to have the highest level of reproducibility for MTrP palpation [21]. After this training, the compression techniques were performed on the dynamometer, asking the participants to maintain a constant amount of pressure, like that exerted during the previous training. The participants were blinded to the data displayed by the dynamometer to prevent feedback.

Finally, the participants assessed the degree of comfort felt in the execution of each technique on an NRS scale. To preserve the blinding of the study, the researcher who performed (ISH) the registry was not the same who performed (ESC) the measurement and analysis of the data. All the participants received the relevant information and signed the informed consent to participate.

### 2.4. Instruments

Dynamometer. A microFET3 wireless muscle dynamometer was used to measure the amount of pressure applied during the technique using the TBS software version 11.0.1 (Hoggan Health Industries, West Jordan, UT, USA, 2000). This device has an intra-tester reliability rated as moderate–excellent [ICC 0.56–0.92] [22] and was calibrated prior to each test performed.

3TOOL^®^ is an ergonomic tool designed to help physiotherapists perform different techniques, used mainly for treatment of MTrPs, soft and fascial tissue [23], which was developed by researchers at the University of Zaragoza in 2013 and it is used in the daily practice of many physiotherapists (see Figure 1).

### 2.5. Outcomes

#### 2.5.1. Primary Outcome

Pressure: The pressure exerted by the physiotherapist was recorded during both modalities of PR for 80 s. The mean pressure exerted during the 80 s and the mean pressure in each 5-s interval were collected. The units of measurement were in Newtons/cm^2^ (N/cm^2^), according to the indications of the International System of Units [24], OldPyM.20 and related studies [14].

#### 2.5.2. Secondary Outcomes

NRS discomfort: The degree of discomfort perceived by the physiotherapist during the technique was evaluated. This used an adapted version of the Numeric Rating Scale for pain (NRS) questionnaire, in which the respondent selects a whole number (0–10) that best reflects the intensity of pain [25]. In this adaptation, higher values represented a higher degree of discomfort.

Existence of (yes/no) previous musculoskeletal pathology in the hand [17,26]: In addition, due to the prevalence of musculoskeletal injuries in the hand in physiotherapists who apply manual therapy techniques, it is interesting to know if the subject suffers or has suffered from any related pathology in order to see the relationship with the NRS comfort variable.

Other sociodemographic variables: such as age, sex, and the number of years of clinical experience of the participating physiotherapists, as these may influence the development of the technique.

### 2.6. Sample Size

A convenience sample of 30 participants was used according to studies carried out previously on healthy individuals with MTrPs in the upper trapezius muscle [27].

### 2.7. Statistical Analysis

The statistical package used for the analysis of the variables was IBM SPSS Statistics v.21 (SPSS, Inc., Chicago, IL, USA) together with Microsoft Excel 2010 (Microsoft, Redmond, WA, USA). Descriptive statistics were performed on all sociodemographic variables showing the data as mean ± standard deviation (SD).

The pressure values exerted by the participants have been analysed globally for the 80-s duration and in 5-s intervals to be able to observe the moment of the technique in which the greatest difference appears in terms of to the constant maintenance of pressure. To compare the pressure exerted in both groups, the final global mean and each 5-s interval for each modality were compared with the Student’s *t*-test. Likewise, the mean pressure in each 5-s interval with respect to 1–5 s interval was compared by applying a Student’s *t*-test for related samples.

The NRS Comfort variable was analyzed using an ANOVA test in which professional experience has been included as a covariate.

The level of statistical significance has been established at a value of *p* < 0.05 and the magnitude of the differences in the tests has also been expressed with the Cohen effect size (ES), the criterion for interpretation being: <0.2 trivial, 0.2 to 0.5 small, 0.5 to 0.8 moderate, and >0.8 large [28].

## 3. Results

A total of 30 volunteers were recruited to participate in the study, 15 were students and the other 15 were physiotherapists/lecturers (see Figure 2), both from the Physiotherapy Degree at Universidad San Jorge (see Table 1).

### 3.1. Variability of Pressure during the Technique

The sequence of the mean pressure exerted for both modalities throughout the duration of the technique is shown in Figure 3. There were significant differences in the mean pressure exerted during the 80 s between the manual and the instrumental compression (*p* = 0.043, ES = 0.25) independently of the users (Table 2), with the most significant differences after 45 s of application to the end (Table 3).

Regarding the intra-group analysis of pressure analyzed at 5-s interval between groups, the results show a significant decrease of pressure (−0.42; *p* = 0.008) in the manual group after the first interval (6–10 s). In the case of the instrumental group, the first significant decrease (−0.30; *p* = 0.013) starts at the second interval (11–15 s) (Table 4).

### 3.2. NRS Discomfort Questionnaire

The PR technique using the 3TOOL(PR-3T) proved to be more comfortable than the PR technique with finger application (PR-Man). The degree of discomfort perceived by students is 4.67 (±1.35) for the PR-Man technique and 1.93 (±0.88) for the PR-3T technique. In the case of professionals, the value obtained for discomfort for PR-Man technique is 4.87 (±2.13) and 3.33 (±1.54) for the PR-3T technique performed. There were statistically significant differences in the degree of discomfort perceived by students and physiotherapists in favor of the PR-3T technique (Figure 4), with an effect size of 1.34.

## 4. Discussion

The results obtained in this study show statistically significant differences in the maintenance of the initial level of pressure exerted with a device when compared to the direct application of pressure with the fingers. Besides, the decrease in the pressure exerted starts before in the manual group (interval 6–10 s) than in the instrumental group (interval 11–15 s). In the analysis by time intervals, a greater decrease in the pressure exerted is observed when compression is made manually, especially after 45 s which increases until the end of the record (80″). Besides, the instrumental application shows a less degree of discomfort perceived by both the students and the physiotherapists, with a large effect size.

Few studies have been carried out to evaluate the pressure in terms of effectiveness of PR techniques. Fryer et al. [14] evaluated the effect of increasing manual pressure monitored by patient perception (“moderate but easily tolerable” pain–value of 7 out of 10) on MTrPs in the upper trapezius with a digital algometer attached to the tip of the palpating thumb (pressure sensor) during all the time that the technique was performed, comparing a therapeutic pressure and a sham pressure (non-therapeutic pressure). In this study there were no significant changes in the PPT following the sham treatment in the control group, whereas there was a significant decrease in the group that received the manual PR technique, which led the authors to conclude that the reduction in tenderness was due to a change in tissue sensitivity, rather than an unintentional reduction of pressure by the examiner. Similar results were found by Taleb et al. [13], who compared manual pressure versus pressure monitored and applied by means of an algometer and found only significant differences on PPT in the algometer group, where the amount of pressure was maintained throughout the technique. Therefore, although a mechanism related to specific compression of the MTrPs may explain differences in pain sensitivity, which could be also related with the consistent compression exerted for the whole treatment.

As we can see in our study the pressure decreases from baseline at both modalities, manual and instrumental, being that pressure loss greater and earlier in the manual application than in the instrumental. These results of our study support the use of instrumental techniques, as they allow a more consistent pressure during the time that the technique is applied, which is a guarantee of good practice by maintaining the constant compression necessary in these techniques. Moreover, it is important to consider that although the scientific evidence shows that both manual compression and dry needling of MTrPs are effective to decrease myofascial pain [29,30], the use of dry needling may have different limitations in the clinical practice such as being painful or not being according to the patient’s preferences. In this case, when manual compression is considered as the preferred treatment option, the therapist may consider the benefits of performing the compression with an instrument. According to our research, instrumental compression has shown to be not only more effective to maintain pressure, which is key when performing a compression technique on a MTrP, but also to be more comfortable for the therapist.

To our knowledge, the capacity to maintain a constant pressure by the physiotherapist without feedback has not been yet evaluated. Different studies have demonstrated the clinical effectiveness of PR techniques when they are performed in an isolated manner [11] or included into a multimodal treatment to improve pain and increase the PPTs [2,11,31], as well as the effectiveness when these techniques are performed with two different tools [18,19]. However, to our knowledge, the differences between PR techniques when they are performed manually or with an ergonomic device has not been analysed and therefore the working mechanisms remained unknown. Regarding the mechanisms involved, it is possible that the decrease of pressure exerted during the application of these techniques may result from a combination of accumulated fatigue and adaptation. However, it is not possible to distinguish the role that fatigue and/or adaptation may play in the decrease of pressure during the PR application, as well as if there are any other factors that may contribute. Moreover, the previous experience performing PR techniques did not show to be an important factor in terms of applying compression more consistently when a device specifically designed for this type of techniques is used.

The comfort level of the instrumental technique perceived by students has been greater than by lecturers (physiotherapists with clinical experience). It could be explained from a lack of training and skill of the students, but it would not be a consistent answer since the effectiveness of technical aids has already been demonstrated when they are used by the patient himself [18,19]. At any case, the greater degree of comfort provided by the tool is not only important for this reason, but also because it may help to prevent injuries to the wrist and hand that sometimes occur in the long term with these types of techniques [31,32].

One of the limitations of the work has been the selection of the sample in which all the participating subjects belonged to the Universidad San Jorge, which impedes the generalization of results. Another limitation is the amount of time evaluated, which was 80 s, a limitation of the dynamometer’s own software whose maximum record is 80 s, whereas most of the described compression techniques last between 60 and 90 s [16]. In future studies, it would be appropriate to evaluate the effectiveness of this kind of tool on real patients. Moreover, it is necessary to consider that this research has been carried out by analyzing the pressure maintenance over a dynamometer, so the results cannot be transferred directly to clinical practice as there are many factors, such as the existence of fat tissue, that were impossible to reproduce.

## 5. Conclusions

The use of a specific and ergonomic device to apply compression during PR techniques on MTrPs proved to be better than manual compression in terms of maintaining a consistent pressure, which becomes more evident after 45 s, independently of previous experience of performing these techniques. Moreover, participants using a device to apply compression experienced a greater degree of comfort during the application, independently of previous clinical experience. Therefore, the sustained compression necessary in manual pressure release techniques in the treatment of MTrPs can be performed with assistive tools that guarantee a uniform compression maintained throughout the application of the technique and are more comfortable for the physiotherapist.

## Figures and Tables

**Figure 1 ijerph-18-12121-f001:**
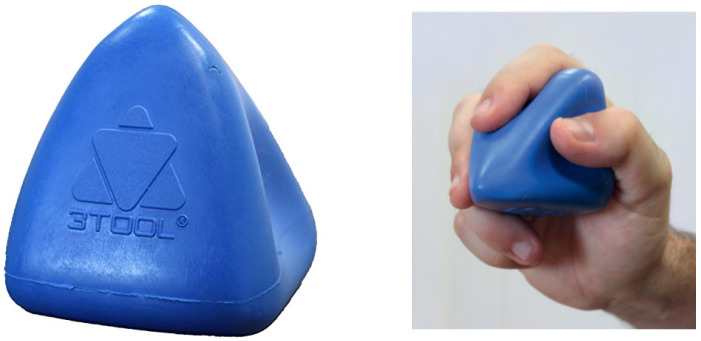
3TOOL^®^.

**Figure 2 ijerph-18-12121-f002:**
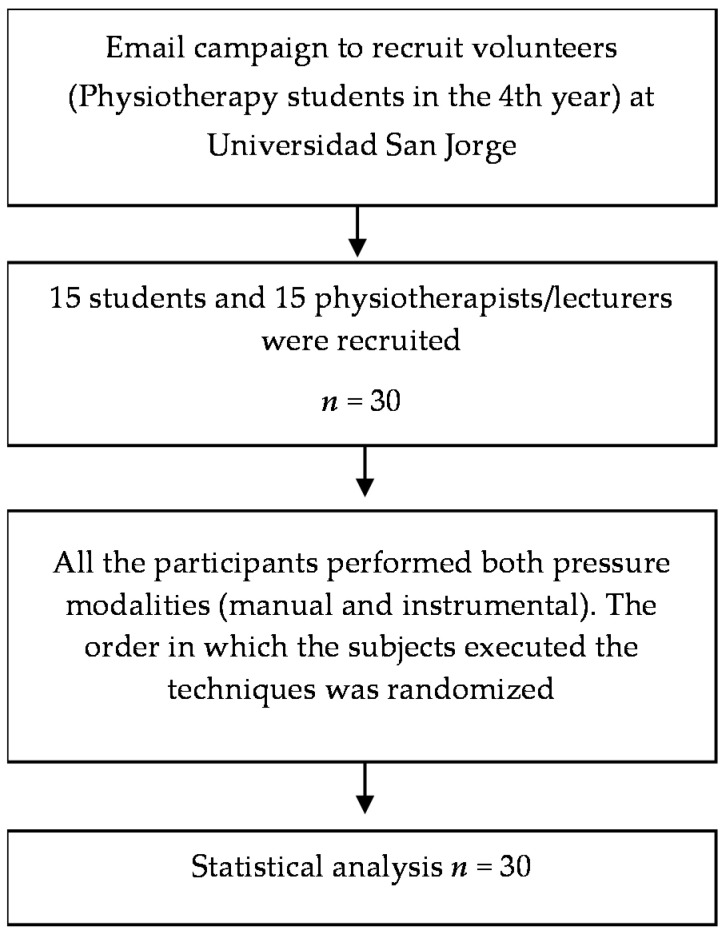
Flow diagram of study.

**Figure 3 ijerph-18-12121-f003:**
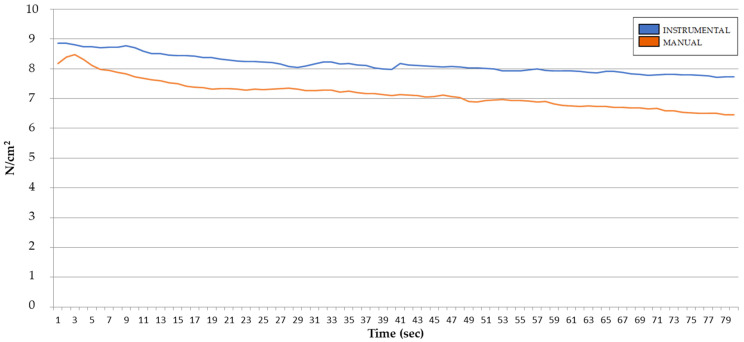
Sequence of pressure exerted throughout the duration of the technique (mean: N/cm^2^/s) with the instrument and manually.

**Figure 4 ijerph-18-12121-f004:**
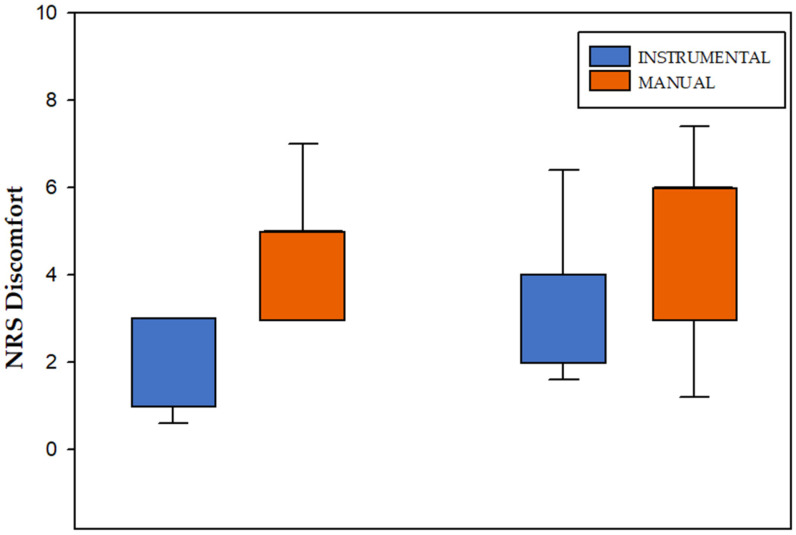
NRS Discomfort questionnaire. NRS: Numeric Rating Scale.

**Table 1 ijerph-18-12121-t001:** Characteristics of participants.

Sample Characteristics	Students (*n* = 15)	Physiotherapists (*n* = 15)	Total (*n* = 30)
Age (SD)	23.50 (3.16)	32.50 (5.73)	28.03 (6.45)
Sex (males/females)	6/9	4/11	10/20
Professional experience (years)	0	10.80 (6.09)	10.80 (6.09)

**Table 2 ijerph-18-12121-t002:** Comparison of the mean global pressure (N/cm^2^).

Interval (s)	Technique	Mean (SD)	95% CI Mean (Inferior; Superior)	*p*-Value	Effect Size (ES)
1–80	Instrumental	8.22 (4.04)	(7.07; 10.5)	0.043 *	0.253
	Manual	7.27 (3.46)	(7.01; 9.59)		

* Statistical significant differences (*p*-value < 0.05).

**Table 3 ijerph-18-12121-t003:** Comparison of pressure exerted (N/cm^2^) at 5-s interval between groups.

Interval (s)	Technique	Mean (SD)	95% IC Mean (Inferior; Superior)	*p*-Value *
1–5	Instrumental	8.80 (4.62)	(7.07; 10.50)	0.361
	Manual	8.30 (3.45)	(7.01; 9.59)	
6–10	Instrumental	8.73 (4.57)	(7.02; 10.40)	0.083
	Manual	7.87 (3.48)	(6.57; 9.17)	
11–15	Instrumental	8.50 (4.33)	(6.88; 10.11)	0.050
	Manual	7.58 (3.41)	(6.31; 8.86)	
16–20	Instrumental	8.39 (4.33)	(6.77; 10.00)	0.033 *
	Manual	7.36 (3.39)	(6.10; 8.62)	
21–25	Instrumental	8.25 (4.25)	(6.66; 9.83)	0.052
	Manual	7.31 (3.46)	(6.02; 8.60)	
26–30	Instrumental	8.12 (4.00)	(6.62; 9.60)	0.084
	Manual	7.31 (3.65)	(5.95; 8.67)	
31–35	Instrumental	8.19 (3.99)	(6.70; 9.68)	0.049 *
	Manual	7.26 (3.60)	(5.91; 8.60)	
36–40	Instrumental	8.05 (3.68)	(6.60; 9.49)	0.064
	Manual	7.16(3.54)	(5.83; 8.48)	
41–45	Instrumental	8.12 (3.94)	(6.64; 9.59)	0.041 *
	Manual	7.09 (±3.42)	(5.81; 8.36)	
46–50	Instrumental	8.05 (3.92)	(6.58; 9.51)	0.032 *
	Manual	7.01 (3.37)	(5.75; 8.25)	
51–55	Instrumental	7.96 (3.90)	(6.50; 9.41)	0.036 *
	Manual	6.95 (3.33)	(5.70; 8.19)	
56–60	Instrumental	7.95 (3.94)	(6.48; 9.42)	0.022 *
	Manual	6.86 (3.28)	(5.63; 8.08)	
61–65	Instrumental	7.90 (3.87)	(6.45; 9.34)	0.016 *
	Manual	6.75 (3.21)	(5.54; 9.94)	
66–70	Instrumental	7.84 (3.89)	(6.39; 9.29)	0.017 *
	Manual	6.69 (3.28)	(5.46; 7.91)	
71–75	Instrumental	7.80 (3.86)	(6.35; 9.24)	0.009 *
	Manual	6.58 (3.25)	(5.36; 7.79)	
76–80	Instrumental	7.74 (3.72)	(6.35; 9.13)	0.005 *
	Manual	6.48 (3.19)	(5.29; 7.67)	

* Statistical significant differences (*p*-value < 0.05).

**Table 4 ijerph-18-12121-t004:** Intra-group differences of pressure exerted (N/cm^2^) at 5-s intervals from baseline.

Interval (s)	6–10		11–15		16–20		21–25		26–30		31–35		36–40		41–45		46–50		51–55		56–60		61–65		66–70		71–75		76–80	
Technique	D	*p*-Value	D	*p*-Value	D	*p*-Value	D	*p*-Value	D	*p*-Value	D	*p*-Value	D	*p*-Value	D	*p*-Value	D	*p*-Value	D	*p*-Value	D	*p*-Value	D	*p*-Value	D	*p*-Value	D	*p*-Value	D	*p*-Value
Instrumental	−0.07	0.067	−0.30	0.013 *	−0.41	0.005 *	−0.55	0.007 *	−0.68	0.001 *	−0.61	0.001 *	−0.75	<0.001 *	−0.68	0.003 *	−0.75	0.003 *	−0.84	0.001 *	−0.85	0.002 *	−0.90	0.001 *	−0.96	0.001 *	−1.00	0.001 *	−1.06	<0.001 *
Manual	−0.42	0.008 *	−0.71	0.001 *	−0.94	<0.001 *	−0.99	<0.001 *	−0.98	0.001 *	−1.04	<0.001 *	−1.14	<0.001 *	−1.21	<0.001 *	−1.29	<0.001 *	−1.35	<0.001 *	−1.44	<0.001 *	−1.55	<0.001 *	−1.61	<0.001 *	−1.72	<0.001 *	−1.81	<0.001 *

D: Difference of each interval from baseline (first interval: 1–5 s) interval. * Statistical significant differences (*p*-value < 0.05).

## Data Availability

Data can be requested to the corresponding author at pherrero@unizar.es.

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
