# Peer review of "Is Instrumental Compression Equally Effective and Comfortable for Physiotherapists and Physiotherapy Students than Manual Compression? A Comparative Cross-Sectional Study"

_ijerph, 2021, doi:10.3390/ijerph182212121_

Round 1

Reviewer 1 Report

Dear authors,

I am glad to have the opportunity to review this manuscript of the effectiveness of pressure, the pain intensity perceived, and the protection of the hands, wrists, and thumbs of the physiotherapists when they use the manual compression techniques compared to an ergonomic tool on the myofascial trigger points therapy.

Specific comments:

Methods

- Describe the relevant dates, including periods of recruitment

- Explain why you conducted the compression technique on the upper-trapezius muscle and not in other muscles regions such as suboccipital, supraspinatus, gluteus, etc.

- Page 3, lines 134-142. Are there any pictures of the 3. TOOL®?

- Explain how the study size was arrived at (software, power, significance level, and mean or standard deviation data employed)

- The prevalence of musculoskeletal pain disorders increases with age and appears to be associated with the body mass index (BMI). So, please provide how you control these confounders in the statistical analysis

Results

- Report numbers of individuals at each stage of study. Consider use of a flow diagram

Discussion

- Discuss why the compression techniques (Chaitow modality, 3.TOOL®) are more effective than dry needling in order to increase the scientific soundness of this article

Author Response

Thanks for reviewing the manuscript and the constructive review. We attach a point by point reply

Reviewer 2 Report

The article is of scientific interest and in line with the aims of the magazine. The auhots guidelines have been revised and the manuscript does not require a revision of the English language by a native speaker. There are some concerns that need to be addressed.

KEYWORDS
In order to increase the visibility of the article, do not use keywords already present in the title.

INTRODUCTION
The introduction adequately describes the background.
The purpose of the study is clearly stated.
"Manual compression techniques are commonly used by physiotherapists in their daily clinical practice, as they have shown to be effective to manage pain provoked by Myofascial Trigger Points (MTrPs) [1-7]." The 7 references are too many and not all consistent or necessary. It would be more correct to add a sentence and divide them, for example: "Manual compression is one of the most used manual therapy techniques in clinical practice by physiotherapists [1-3]. This method has proved to be effective in the treatment of numerous scjeletal muscle dysfunctions such as for example Myofascial Trigger Points (MTrPs) [4-7]. Consider adding the following reference:
- Bernetti A, Agostini F, de Sire A, Mangone M, Tognolo L, Di Cesare A, Ruiu P, Paolucci T, Invernizzi M, Paoloni M. Neuropathic Pain and Rehabilitation: A Systematic Review of International Guidelines. Diagnostics (Basel). 2021 Jan 5; 11 (1): 74. doi: 10.3390 / diagnostics11010074.
"demonstrated to be key for the effectiveness in decreasing pain" serves reference. Consider and add as a reference:
-Paolucci T, Bernetti A, Paoloni M, Capobianco SV, Bai AV, Lai C, Pierro L, Rotundi M, Damiani C, Santilli V, Agostini F, Mangone M. Therapeutic Alliance in a Single Versus Group Rehabilitative Setting After Breast Cancer Surgery : Psychological Profile and Performance Rehabilitation. Biores Open Access. 2019 Jul 3; 8 (1): 101-110. doi: 10.1089 / biores.2019.0011.

MATERIALS AND METHODS
The inclusion and exclusion criteria should be better described.
The interventions carried out are clear and suitable for the purpose of the study.
The outcomes are consistent with the purpose of the study.
I think we need a figure of the instruments used.

RESULTS DISCUSSION
The results are fluidly written and well argued in the discussion section.

TABLES
The tables are clear and adequately complement the text.

FIGURES
The frames are of good quality.

REFERENCES
The references are recent and relevant to the topic.

Author Response

(The authors gave the same response as above.)

Reviewer 3 Report

Thank you for the opportunity to review this manuscript. In order to improve the understanding of the manuscript. Some points should be discussed. Are they:

- Please define the objective better. The objective needs to be better structured.
- The study was carried out with a specific sample. Coming from a specific place. This can be better described in the title and objective.
- Methodology needs to be better described. Mainly regarding the procedures performed.
- The description of the sample calculation is limited. It needs to be better defined.
- In the discussion topic, better define the clinical applicability of the results.
- The conclusion must respond to the objective. The conclusion needs to be better written.

Author Response

(The authors gave the same response as above.)

Round 2

Reviewer 1 Report

Dear authors,

I appreciate the possibility to review the revised version of this manuscript. The authors have improved the quality of the manuscript with your corrections.